

# DNAJ heat shock protein family member C1 can regulate proliferation and migration in hepatocellular carcinoma

Yu-Chun Fan[1,2,3], Zhi-Yong Meng[4], Chao-Sheng Zhang[2], De-Wei Wei[5], Wan-Shuo Wei[2], Xian-Dong Xie[2], Ming-Lu Huang[5] and Li-He Jiang[1,2,3,6]

[1] Medical College, Guangxi University, Nanning, Guangxi, China
[2] School of Basic Medical Sciences, Youjiang Medical University for Nationalities, Baise, China
[3] Key Laboratory of Minimally Invasive Techniques & Rapid Rehabilitation of Digestive System Tumor of Zhejiang Province, Zhejiang, China
[4] First Clinical Medical College, Guangxi Traditional Chinese Medical University, Nanning, China
[5] School of Stomatology, Youjiang Medical University for Nationalities, Baise, China
[6] Key Laboratory of Cellular Physiology (Shanxi Medical University), Ministry of Education, Shanxi, China

Corresponding author
Li-He Jiang, jianglihe@ymun.edu.cn

## ABSTRACT

**Background**. DNAJ heat shock protein family (Hsp40) member C1(DNAJC1) is a member of the DNAJ family. Some members of the DNAJ gene family had oncogenic properties in many cancers. However, the role of DNAJC1 in hepatocellular carcinoma (HCC) was unclear.

**Methods**. In this study, expression and prognostic value of DNAJC1 in HCC were analyzed by bioinformatics. Quantitative real-time PCR and Western blotting were used to verify DNAJC1 expression in liver cancer cell lines. Furthermore, immunohistochemical (IHC) was used to detect DNAJC1 expression in liver cancer tissues. Subsequently, the effect of DNAJC1 on the proliferation, migration, invasion and apoptosis of HCC cells was detected by knocking down DNAJC1. Finally, gene set enrichment analysis (GSEA) was used to investigate the potential mechanism of DNAJC1 and was verified by Western blotting.

**Results**. DNAJC1 was highly expressed in HCC and was significantly associated with the prognosis of patients with HCC. Importantly, the proliferation, migration and invasion of Huh7 and MHCC97H cells were inhibited by the knockdown of DNAJC1 and the knockdown of DNAJC1 promoted Huh7 and MHCC97H cell apoptosis. Furthermore, compared to the negative control group, DNAJC1 knockdown in Huh7 and MHCC97H cells promoted the expression of p21, p53, p-p53(Ser20), Bax and E-cadherin proteins, while inhibiting the expression of PARP, MMP9, Vimentin, Snai1, Bcl-2 and N-cadherin proteins.

**Conclusions**. DNAJC1 had a predictive value for the prognosis of HCC. Knockdown of DNAJC1 may inhibit HCC cell proliferation, migration and invasion and promote the HCC cell apoptosis through p53 and EMT signaling pathways.

## INTRODUCTION

Hepatocellular carcinoma (HCC) is one of the primary malignant tumors with high mortality in the world (*Li et al., 2020*; *Ouyang et al., 2022*). Currently, the treatment strategies for patients with HCC mainly include surgical resection, chemotherapy, and radiation therapy (*Heimbach et al., 2018*). However, most patients had already lost surgical treatment in the advanced stage when diagnosed with HCC. Therefore, it is important to find new therapeutic and prognostic targets for HCC in which it is easier to realize personalized treatment and minimally invasive or even non-invasive treatment (*Craig et al., 2020*).

DNAJ heat shock protein family (HSP40) is an evolutionarily conserved protein family, and more than 40 members of the HSP40 family have been identified, whose functions include regulating transcriptional translation and protein folding (*Fan, Lee & Cyr, 2003*; *Qiu et al., 2006*; *Daugaard, Rohde & Jäättelä, 2007*; *Vos et al., 2008*). DNAJ proteins are differentially expressed in human tissues and have been shown to promote or inhibit cancer (*Sterrenberg, Blatch & Edkins, 2011*). DNAJC1 is a member C1 of the DNAJ family of heat shock proteins. The membrane protein encoded by DNAJC1 is a heat shock protein similar to DNAJ, which can bind to the molecular chaperone BiP (*Kroczynska et al., 2005*). Currently, some members of the DNAJ gene family have been studied in tumors. For example, DNAJB6 has an effect on iron sagging in esophageal squamous cell carcinoma (*Jiang et al., 2020*). The high expression of DNAJC12 in gastric cancer affects gastric cancer invasion (*Uno et al., 2019*). DNAJB4 is a novel biomarker for breast cancer (*Acun et al., 2017*). Tazarotene-induced gene 1 interacts with DNAJC8 and regulates glycolysis in cervical cancer cells (*Wang et al., 2018*). DNAJB8 is considered a novel immunotherapy target for colon cancer cells (*Morita et al., 2014*). Also, DNAJC21 mutations may be associated with cancer-prone bone marrow failure syndrome (*Tummala et al., 2016*). However, the role of DNAJC1 in HCC is still unclear.

We hypothesized that DNAJC1 was important in the occurrence and development of HCC. In this study, we elucidated the value of DNAJC1 in the diagnosis and prognosis of HCC by bioinformatic analysis and experimental verification. Our data suggested that DNAJC1 had a predictive value for the prognosis of HCC and can affect the proliferation, migration, invasion and apoptosis of HCC cells through p53 and EMT signaling pathways.

## MATERIAL AND METHODS

### Data collection

Expression data and clinical data were downloaded from The Cancer Genome Atlas database (TCGA, https://cancergenome.nih.gov/)(374 tumor samples and 50 normal samples) (*Cancer Genome Atlas Research Network, 2013*). The TCGA database is open to the public, and all data have been agreed to be used for analysis and have obtained moral recognition. This study is based on open source data, strictly abides by the release guidelines and access policies of the database, and is not bound by other ethics.

## DNAJC1 expression analysis

The Tumor Immune Estimate Resource (TIMER, https://cistrome.shinyapps.io/TIMER/) database (*Li et al., 2017*) was used to compare the expression level of DNAJC1 in all TCGA tumors. Subsequently, DNAJC1 mRNA expression analysis was based on TCGA database by R package "Limma", and the "ggplot2" package was used to draw the box plot for visualization.

## Prognostic role of DNAJC1 in HCC

The UALCAN database (https://ualcan.path.uab.edu/) (*Chandrashekar et al., 2017*) was a website tool for online analysis and extraction based on the clinical data of 31 cancer types in TCGA database. In this study, the correlation between DNAJC1 expression and the prognosis of patients was explored by the UALCAN database.

## Gene set enrichment analysis (GSEA)

Gene set enrichment analysis (GSEA) (*Subramanian et al., 2005*) was used to explore the related signaling pathways. In this study, TCGA samples were divided into high-low expression groups according to the median expression of DNAJC1. GSEA was performed with the package "ClusterProfiler" (*Yu et al., 2012*) to investigate whether DNAJC1 in the two groups was rich in meaningful tumor hallmarks (h.all.v7.5.symbols.gmt (Hallmarks)). The $p$-value $< 0.05$ and *FDR* $< 0.25$ were considered significantly enriched.

## Immunohistochemistry

Immunohistochemical (IHC) was performed as previously described. We collected 20 pairs of HCC samples and adjacent non-tumor tissues from October 2020 to April 2021 at the Affiliated Hospital of Youjiang Medical University for Nationalities. All procedures were performed according to the Ethics Guidelines for Human Genome/Gene Research and approved by the Ethics Committee of the Affiliated Hospital of Youjiang Medical College for Nationalities (2022090501). Furthermore, we received informed consent from participants in our study. Immunohistochemistry (IHC) was performed on the 20 samples to detect the expression of the DNAJC1 protein.

## Cell culture

The human normal liver cell line ($LO_2$) and three human HCC cell lines (Huh7, MHCC97H and $HepG_2$) were purchased from the Chinese Academy of Sciences. $LO_2$, Huh7, MHCC97H and $HepG_2$ cells were cultured in DMEM (Dulbecco minimum Essential medium) containing 10% FBS (Fetal Bovine Serum). $LO_2$, Huh7, MHCC97H and HepG2 cells were cultured at 37 °C with 5% $CO_2$.

## RNA extraction and quantitative real-time PCR

Total RNA was extracted from cell lines using RNA isolation kit (Axygen, Suzhou, China). The total RNA was then reverse transcribed to cDNA. qRT-PCR was performed with the SYBR Green kit, and it was performed using a thermocycler (Bio-Rad, Hercules, CA, USA) at 95 °C for 600 s, 95 °C for 10 s,65 °C for 60 s, 97 °C for 1 s and 37 °C for 30 s, for a total of 40 cycles. The GAPDH was used as internal control, $2^{-\Delta\Delta ct}$ method was used to quantitatively analyze the expression level of DNAJC1.The sequence of primers used in the

experiment was as follows: DNAJC1 (forward:5′-TTCTCACAGTGGGTCATTATGC-3′; reverse:5′-ACCGAGTTTTGATACATCCACAC -3′); GAPDH (forward: 5′-GGACCTGACCTGCCGTCTAG-3′; reverse:5′-GTAGCCCAGGATGCCCTTGA-3′).

## Western blotting

Total protein was extracted from cells with RIPA lysis buffer solution and protein concentration was determined by the BCA protein determination kit. Proteins were separated by SDS-PAGE and transferred to a polyvinylidene fluoride membrane. The membranes were then sealed with 5% skim milk at room temperature for 1 h, and the membranes combined with the DNAJC1 and GAPDH primary antibody were incubated at 4 °C overnight. The second antibody was incubated for 2 h at room temperature. Protein bands were detected using the ECL chemiluminescence system.

## Transfection

Based on DNAJC1 expression in three HCC cell lines, Huh7 and MHCC97H cell lines were selected for transfection. Specific small interfering RNA (siRNA) targeting DNAJC1 and negative control (NC) siRNA were as follows:

siDNAJC1 (forward:5′-GCCAAGCAACUGAAGGAUUTT -3′; reverse:5′-AAUCCUUC AGUUGCUUGGCTT -3′);

Negative Control (forward:5′- UUCUCCGAACGUGUCACGUTT-3′; reverse:5′-ACGUGACACGUUCGGAGAATT-3′).

The vectors and siRNA were transfected into Huh7 and MHCC97H cell lines using Lipofectamine 3000 reagents. The transfection efficiency was evaluated by qRT-PCR and Western blotting, respectively.

## Cell counting kit-8 (CCK-8) assay

The transfected cells ($5 \times 10^3$/well) were evenly distributed in the 96-well plate with three replicate wells. After transfecting for 24 h, 48 h, 72 h, and 96 h, the 10 μL volume CCK-8 reagent (Dojindo Laboratories, Kumamoto, Japan) was added to each well. The 96-well plate was cultured in an incubator for 1 h, and then its absorbance value at 450 nm was detected in a microplate reader.

## Colony formation assay

$1.0 \times 10^3$ transfected cells were distributed into 6-well plates. After 14 days of incubation, $1.0 \times 10^3$ treated cells were grown into visible colonies. Finally, methanol was used to fix cell colonies and crystal violet was used to stain cell colonies.

## Wound healing assay

Cells were seeded, and scratch wounds were made when cell confluence reached 80%–90% after 48 h transfection. The 10 μL pipette tip was used to draw a straight line perpendicular to the bottom of the 6-well plates. The scratches were cleaned 3 times with PBS buffer. The scratches were calculated after 0 and 24 h incubation at room temperature. Finally, the remaining cells were cultured in a medium free of fetal bovine serum. The pictures were taken under the microscope at 0 and 24 h.

## Transwell migration and invasion assays

For the transwell migration assay, transfected cells were diluted to $1 \times 10^5$ /mL with serum-free RPMI-DMEM medium, and 200 $\mu$L cell suspension was added to the upper transwell chamber and 600 $\mu$L medium containing 20% fetal bovine serum was added to the lower chamber, respectively. For invasion assays, the matrigel was diluted to 1:9 with serum-free DMEM. The transwell chambers were then solidified at 37 °C for 2 h. Then, the transwell invasion experiment was carried out on the basis of the transwell migration experiment. Cells were seeded into a matrigel-coated transwell chamber. After 24 h, cells crossed the inserts were stained with crystal violet and counted under phase contrast microscopy.

## Hoechst 33342 staining

Apoptosis of Huh7 and MHCC97H cells was also evaluated using Hoechst 33342 reagents (Beyotime Biotechnology, Shanghai, China) according to the manufacturer's protocols. The transfected cells were seeded in a 6 well plate and stained with 1 mL Hoechst 33342 dye by incubation in the culture solution for 30 min, and then washed twice with PBS, observed with a fluorescence microscope.

## Flow cytometry

Transfected cells were harvested after transfection 48 h and stained with Annexin V-FITC/PI (BD Biosciences) according to the manufacturer's instructions. Cells were resuspended in 500 μL of experimental buffer, 5 μL of Annexin V-FITC and 5 μL of 100× PI. Cells were incubated at room temperature for 30min in the dark. Finally, the cells were analyzed with flow cytometry.

## Statistical analysis

SPSS 25.0 (SPSS Inc., Chicago, IL, USA) and R 4.04 software (*R Core Team, 2021*) were used for statistical analysis. The two-tailed Student's $t$-test was used for the comparison of samples between two groups, and one-way ANOVA was used for the comparison of samples from more than two groups. All results were presented as mean ± standard deviation (SD). $P < 0.05$ was considered statistically significant. *$P < 0.05$, **$P < 0.01$, ***$P < 0.001$.

# RESULTS

## Expression levels of DNAJC1 in HCC

According to TCGA pan-cancer RNA-seq data, DNAJC1 mRNA expression levels in tumor and normal tissues were analyzed and the results were shown in Fig. 1A. The level of DNAJC1 mRNA expression in breast carcinoma (BRCA), cholangiocarcinoma (CHOL), head and neck squamous cell carcinoma (HNSC), clear cell carcinoma of the kidney (KIRC), papillary cell carcinoma of the kidney (KIRP), liver cancer (LIHC), gastric adenocarcinoma (STAD) and other tumor tissues was significantly higher than in normal tissues ($P < 0.001$). These results suggested that DNAJC1 was up-regulated in most malignant tumors (including HCC) and may be involved in the occurrence and development of cancer.

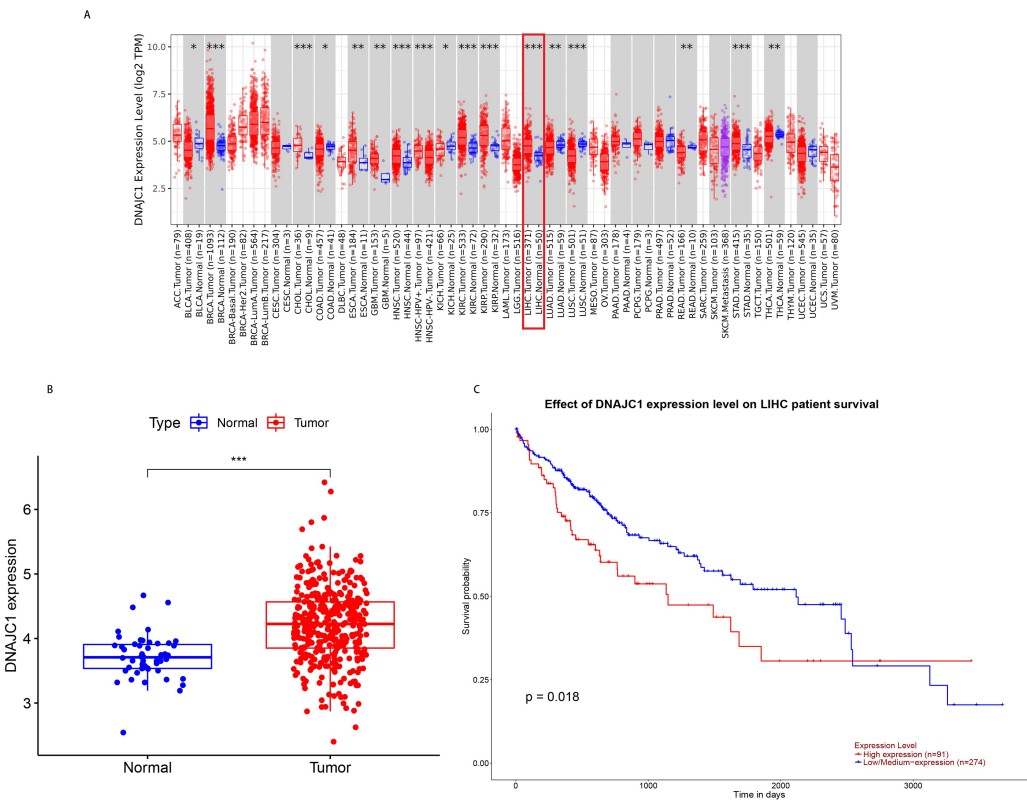

**Figure 1** **The expression and prognostic value of DNAJC1 in HCC patients.** (A) Pan-cancer analysis of DNAJC1 expression in tumor and normal tissues. (B) The expression levels of DNAJC1 in HCC tumor and normal tissues. (C) The survival curves of DNAJC1 in HCC. *: $p < 0.05$; **: $p < 0.01$; ***: $p < 0.001$.

In particular, based on TCGA database, we observed that the expression of DNAJC1 in HCC tissues was significantly higher than in normal liver tissues ($P < 0.001$; Fig. 1B). This result suggested that DNAJC1 may play an important role in HCC.

## Prognostic role of DNAJC1 in HCC

Based on the UALCAN database, the survival curves revealed that HCC patients with high expression of DNAJC1 had higher mortality ($P < 0.05$; Fig. 1C). This result indicated that DNAJC1 was a good prognostic biomarker for HCC patients.

## Validation of DNAJC1 expression levels

qRT-PCR and Western blotting confirmed that DNAJC1 mRNA and protein expression were up-regulated in liver cancer cell lines compared to normal liver cells ($P < 0.05$) (Figs. 2A–2C). Furthermore, DNAJC1 expression was higher in HCC tissues than in adjacent non-tumor tissues by IHC (Fig. 2D). These results suggested that DNAJC1 up-regulation may be closely related to the biological characteristics of HCC.

## Expression of DNAJC1 mRNA and protein after siRNA transfection

We transfected DNAJC1 siRNA into Huh7 and MHCC97H cells, and the transfection efficiency was evaluated by qRT-PCR and Western blotting, respectively. The results

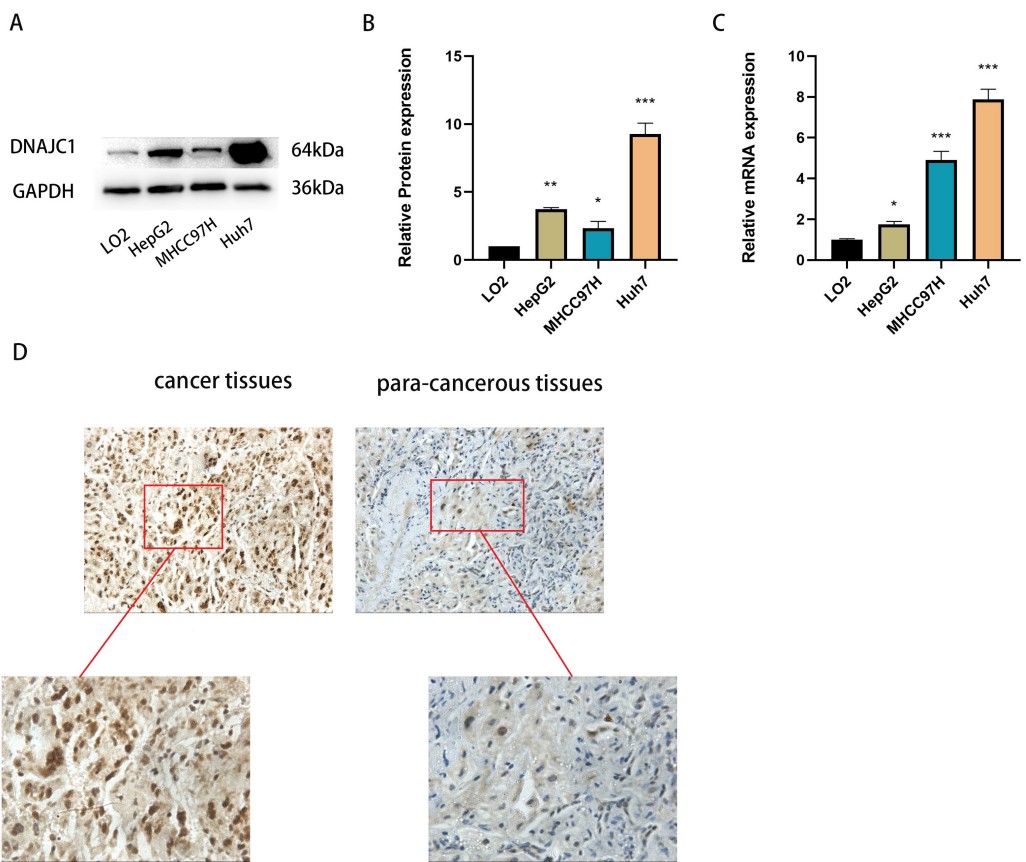

**Figure 2** **The expression level of DNAJC1 in liver cancer cell lines and tissues.** (A–B) The relative protein levels of DNAJC1 in liver cancer cell lines compared with LO2 (normal liver cells). (C) The mRNA expression of DNAJC1 in liver cancer cell lines compared with LO2 (normal liver cells). (D) The protein expression of DNAJC1 in HCC by IHC. *: $p < 0.05$; **: $p < 0.01$; ***: $p < 0.001$.

showed that the mRNA and protein levels of the siDNAJC1 group were significantly lower than those of the NC group ($P < 0.05$) (Figs. 3A–3B).

## Knockdown of DNAJC1 reduces the proliferation of HCC cells

The CCK-8 assay showed that the Huh7 and MHCC97H cell inhibition rate of the siDNAJC1 group increased at 24 h, 48 h, 72 h and 96 h compared to the NC group ($P < 0.05$) (Fig. 4A). Colony formation assay showed that the number of Huh7 and MHCC97H cell clones in the siDNAJC1 group were significantly reduced compared with the NC group ($P < 0.05$) (Fig. 4B). These results showed that DNAJC1 knockdown could inhibit HCC cell proliferation.

## DNAJC1 knockdown promotes HCC cell apoptosis

Hoechst 33342 staining showed that apoptotic cells in the siDNAJC1 group increased significantly and apoptotic cells showed obvious pyknotic deformation and luminous phenomena (Fig. 5A). In addition, flow cytometry showed that the apoptosis rate of NC

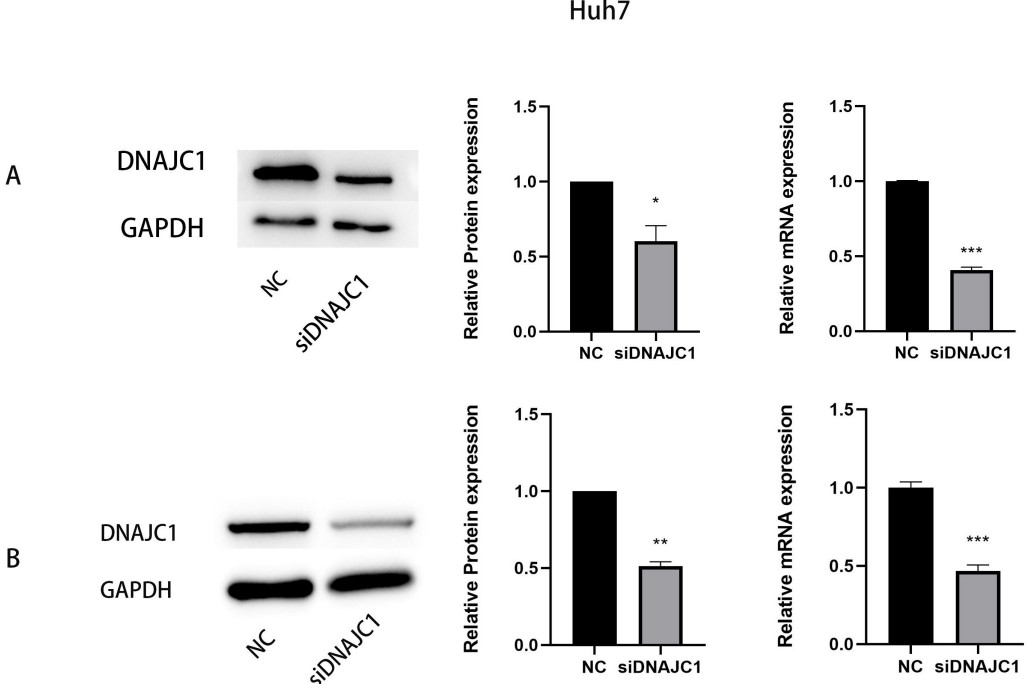

**Figure 3** **The expression of DNAJC1 mRNA and protein after siRNA transfection.** (A) The relative mRNA and protein expression of DNAJC1 in Huh7 cell lines following the transfection of siRNA. (B) The relative mRNA and protein expression of DNAJC1 in MHCC97H cell lines following the transfection of siRNA. *: $p < 0.05$; **: $p < 0.01$; ***: $p < 0.001$.

group was significantly lower than that of siDNAJC1 group ($P < 0.05$) (Fig. 5B). These results showed that DNAJC1 knockdown could promote HCC cell apoptosis.

### DNAJC1 may mediate HCC cell apoptosis *via* p53 signaling pathway

In the experiments mentioned above, we confirmed that DNAJC1 can affect the biological behavior of HCC cells, but the underlying mechanism remained unclear. The GSEA showed that the high expression of DNAJC1 was related to the p53 signaling pathway (Fig. 6A). To investigate the mechanism of HCC cell apoptosis induced by DNAJC1 knockdown, PARP, Bax, Bcl-2, p21, p53 and p-p53 (Ser20) proteins were evaluated. These proteins were essential for HCC cell apoptosis and were important for the p53 signaling pathway. The results showed that compared to the control group, the expression levels of p21, p53, p-p53 (Ser20) and Bax increased significantly, and the expression levels of Bcl-2 and PARP decreased significantly after DNAJC1 knockdown ($P < 0.05$) (Figs. 6B–6C). The results indicated that DNAJC1 may mediate HCC cell apoptosis *via* p53 signaling pathway.

### Knockdown of DNAJC1 reduces the migration and invasion of HCC cells

The results of the wound healing experiment showed that the scratch width of the Huh7 and MHCC97H cells was slightly changed in the siDNAJC1 group, while the scratch width of the Huh7 and MHCC97H cells was significantly reduced in the NC group ($P < 0.05$)

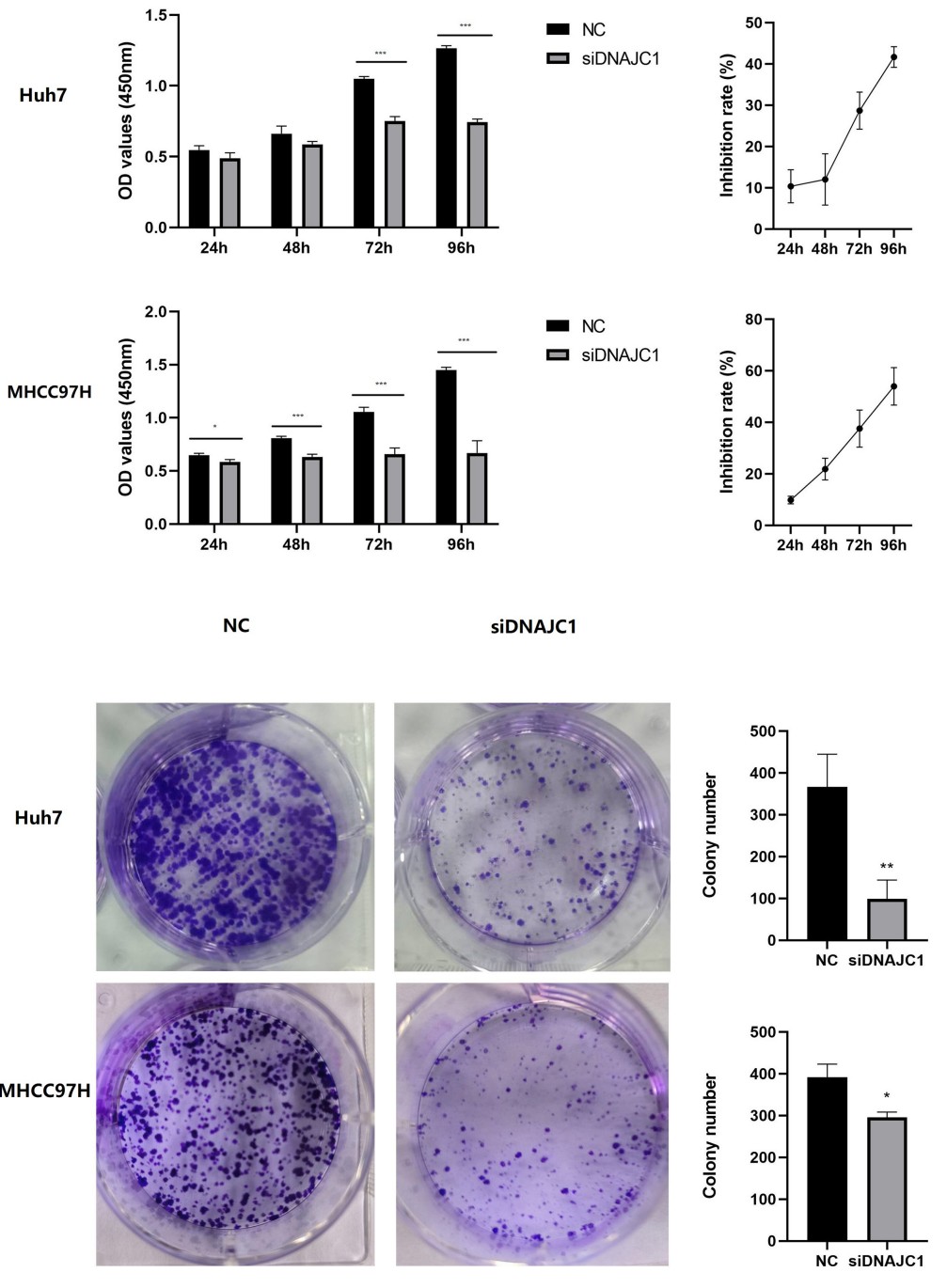

**Figure 4 Knockdown of DNAJC1 reduces the proliferation of HCC cells.** (A) Cell viability assessed by CCK8 assay. (B) Cell viability assessed by clonogenic survival assay. *: $p < 0.05$; **: $p < 0.01$; ***: $p < 0.001$.

(Fig. 7). Transwell migration assay showed that the knockdown of DNAJC1 expression reduced the migration ability of Huh7 and MHCC97H cells ($P < 0.05$) (Figs. 8A–8B). Furthermore, the transwell invasion assay showed that DNAJC1 expression suppression

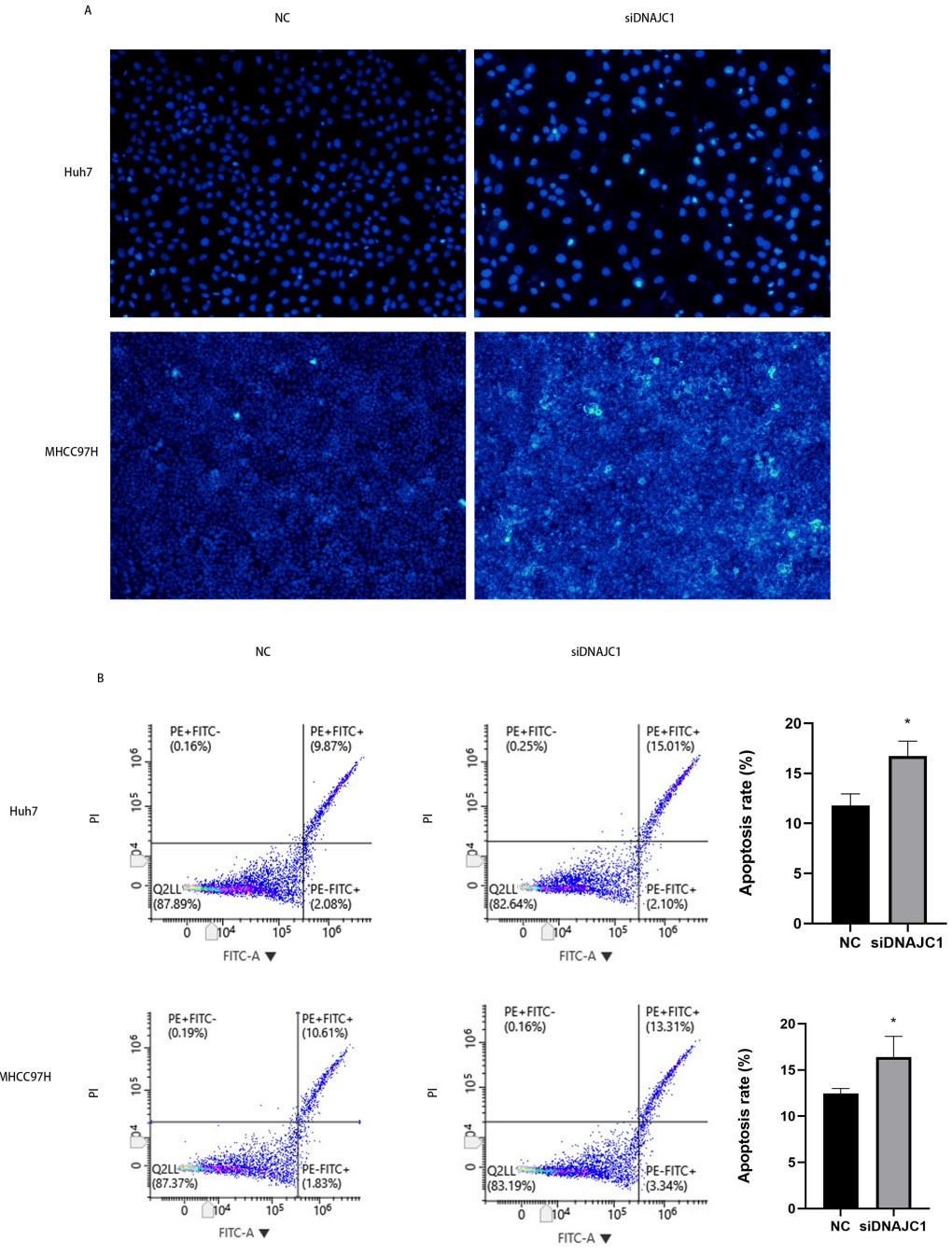

**Figure 5   Knockdown of DNAJC1 promotes the apoptosis of HCC cells.** (A) Silencing DNAJC1 promotes cell apoptosis in Huh7 and MHCC97H cells by Hoechst 33342 assay. (B) Silencing DNAJC1 promotes cell apoptosis in Huh7 and MHCC97H cells by flow cytometry.

inhibited the invasion ability of Huh7 and MHCC97H cells ($P < 0.05$) (Figs. 8C–8D). These results showed that knockdown of DNAJC1 could inhibit the migration and invasion of HCC cells.

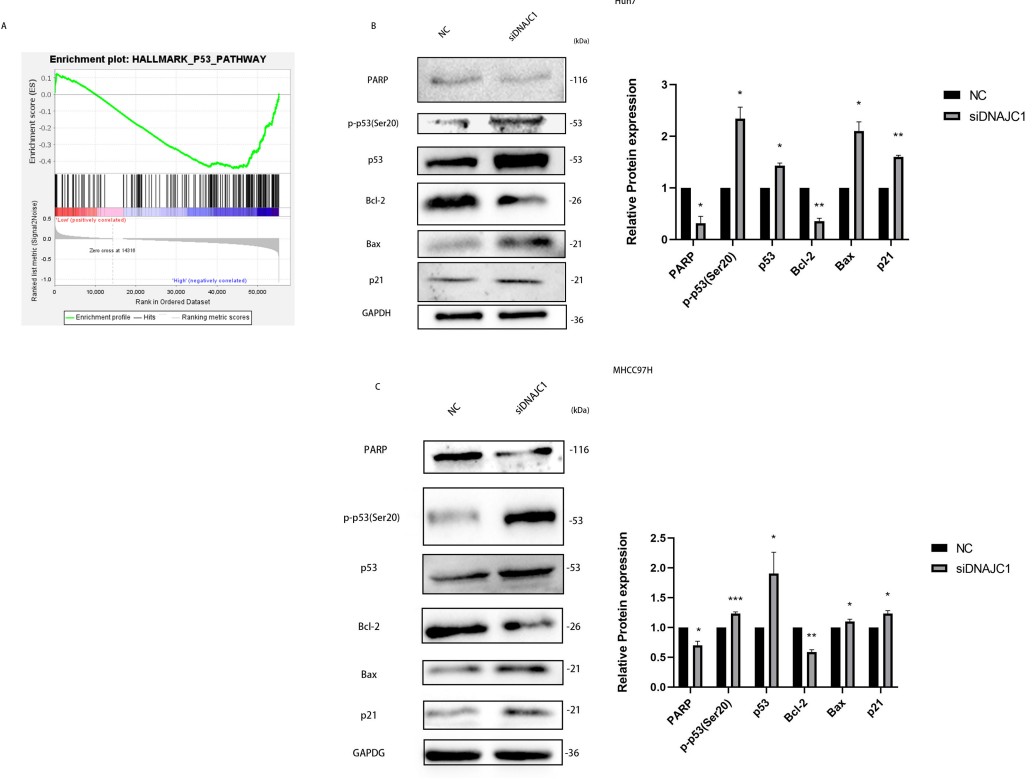

**Figure 6 Silencing DNAJC1 promotes Huh7 and MHCC97H cells apoptosis *via* p53 pathway.** (A) Gene set enrichment analysis. (B–C) Silencing DNAJC1 promotes Huh7 and MHCC97H cells apoptosis *via* p53 pathway. *: $p < 0.05$; **: $p < 0.01$; ***: $p < 0.001$.

## DNAJC1 may mediate HCC cell migration through the EMT signaling pathway

To investigate the mechanism of HCC cell migration induced by DNAJC1 knockdown, E-cadherin, N-cadherin, MMP9, Vimentin, and Snai1 proteins were evaluated. These proteins were essential for the migration of HCC cells, and they were important for EMT signaling pathway. The results showed that compared to the control group, the expression level of E-cadherin increased significantly, and the expression levels of N-cadherin, MMP9, Vimentin, and Snai1 decreased significantly after DNAJC1 knockdown ($P < 0.05$) (Figs. 9A–9B). The results indicated that DNAJC1 may mediate HCC cell migration *via* EMT signaling pathway.

## DISCUSSION

HCC is one of the leading causes of cancer mortality in the word (*Wang et al., 2014*; *Hu et al., 2019*; *Ma et al., 2020*; *Zhao et al., 2020*). Although there are many strategies to treat patients with HCC, the prognosis of patients with HCC is poor (*Jasirwan et al., 2020*; *Wang et al., 2020*; *Wei et al., 2019*). Therefore, it is important to find new targets for early diagnosis and treatment of HCC. Given that DNAJ proteins were differentially expressed

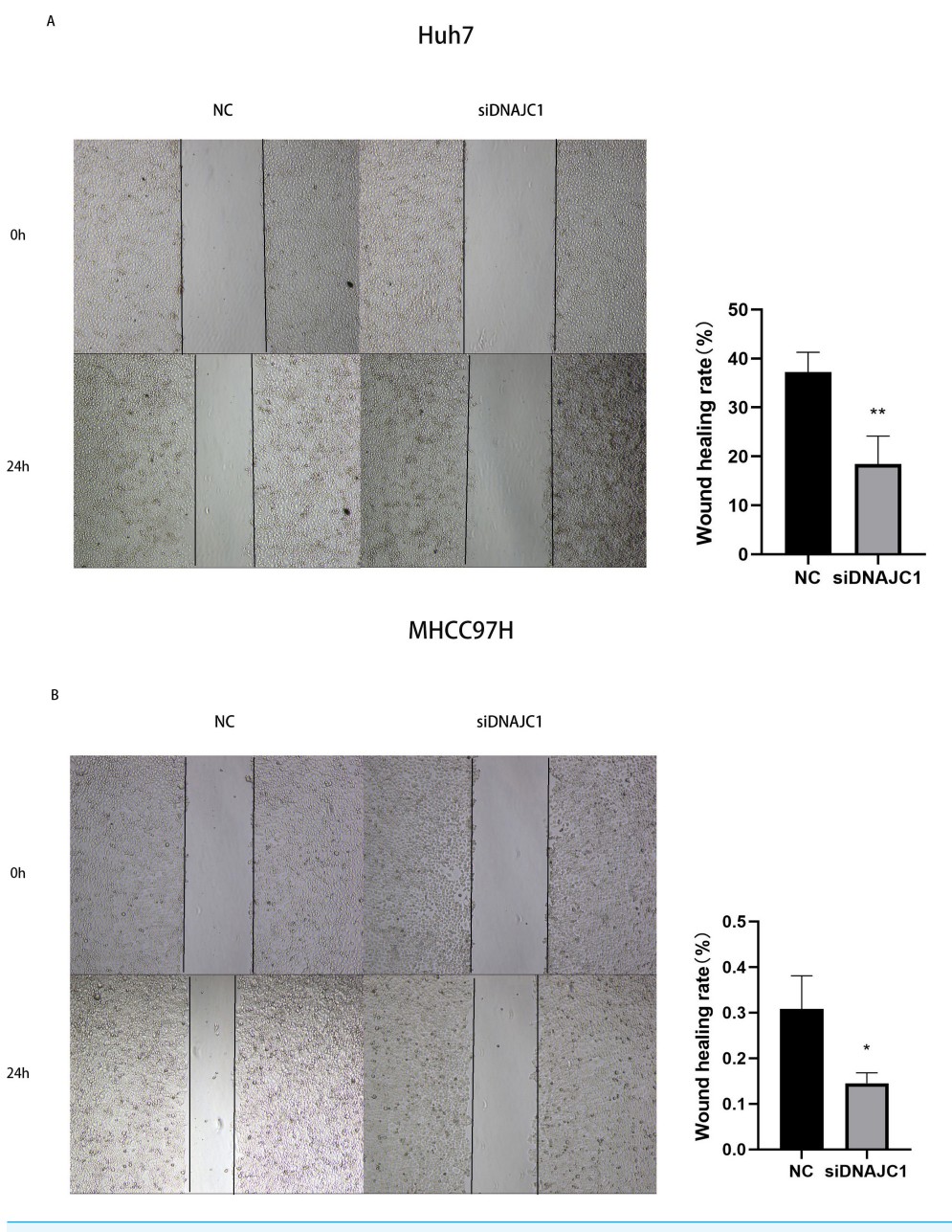

**Figure 7** **Knockdown of DNAJC1 reduces the migration of HCC cells by wound healing assay.** (A) Knockdown of DNAJC1 reduces the migration of Huh7 cells. (B) Knockdown of DNAJC1 reduces the migration of MHCC97H cells. *: $p < 0.05$; **: $p < 0.01$; ***: $p < 0.001$.

in human tissues and had been shown to promote or inhibit cancer (*Li et al., 2019*; *Kim et al., 2019*), we analyzed the biological function of DNAJC1 in HCC.

In this study, pan-cancer analysis showed that DNAJC1 was remarkably elevated in most tumor tissues (including HCC). In addition, our results demonstrated that DNAJC1 could affect the prognosis of HCC patients and may act as an oncogene and participate in the development and progression of HCC.

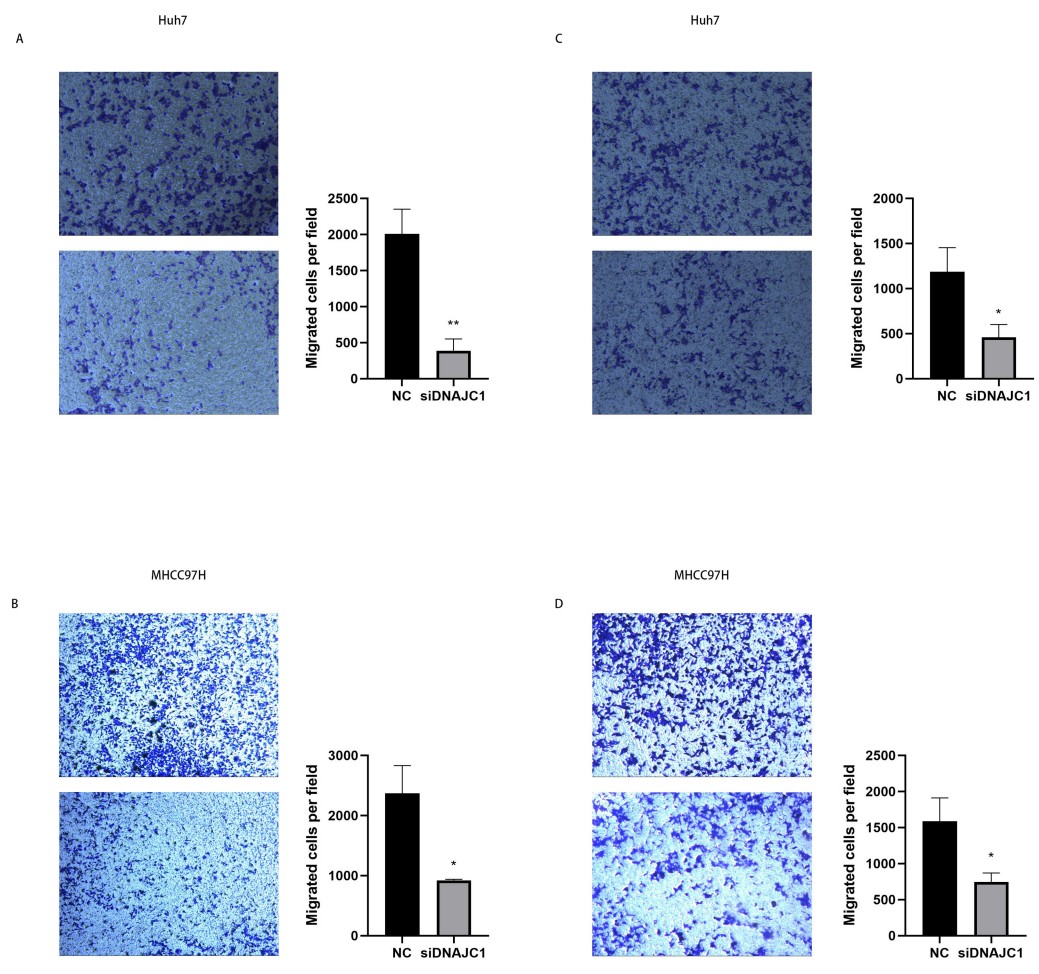

**Figure 8  DNAJC1 inhibited the migration and invasion of HCC cells.** (A–B) Cell migration determined by transwell migration assay. (C–D) Cell invasion determined by transwell invasion assay. *: $p < 0.05$; **: $p < 0.01$; ***: $p < 0.001$.

Furthermore, Western blotting, qRT-PCR and IHC were performed to test the expression level of DNAJC1 in liver cancer cell lines and tissues. We found that DNAJC1 was upregulated in HCC. Finally, we conducted subsequent experiments by knocking down DNAJC1. The results showed that DNAJC1 knockdown could inhibit the proliferation, migration, and invasion of HCC cells, and DNAJC1 knockdown could promote the apoptosis of HCC cells. These results confirmed that DNAJC1 has an effect on the biological behavior of HCC, and targeting DNAJC1 may be a new method to control HCC in clinical practice in the future.

We found that the high expression of DNAJC1 was related to the p53 pathway by GSEA. Tumor suppressor p53 plays a key role in tumor suppression (*Kastenhuber & Lowe, 2017*; *Vousden & Prives, 2009*; *Liu et al., 2015*; *Levine, 2019*). As a transcription factor, p53 mainly regulates multiple target genes through selective transcription and regulates various basic cell responses such as apoptosis, cell cycle arrest, senescence, DNA repair,

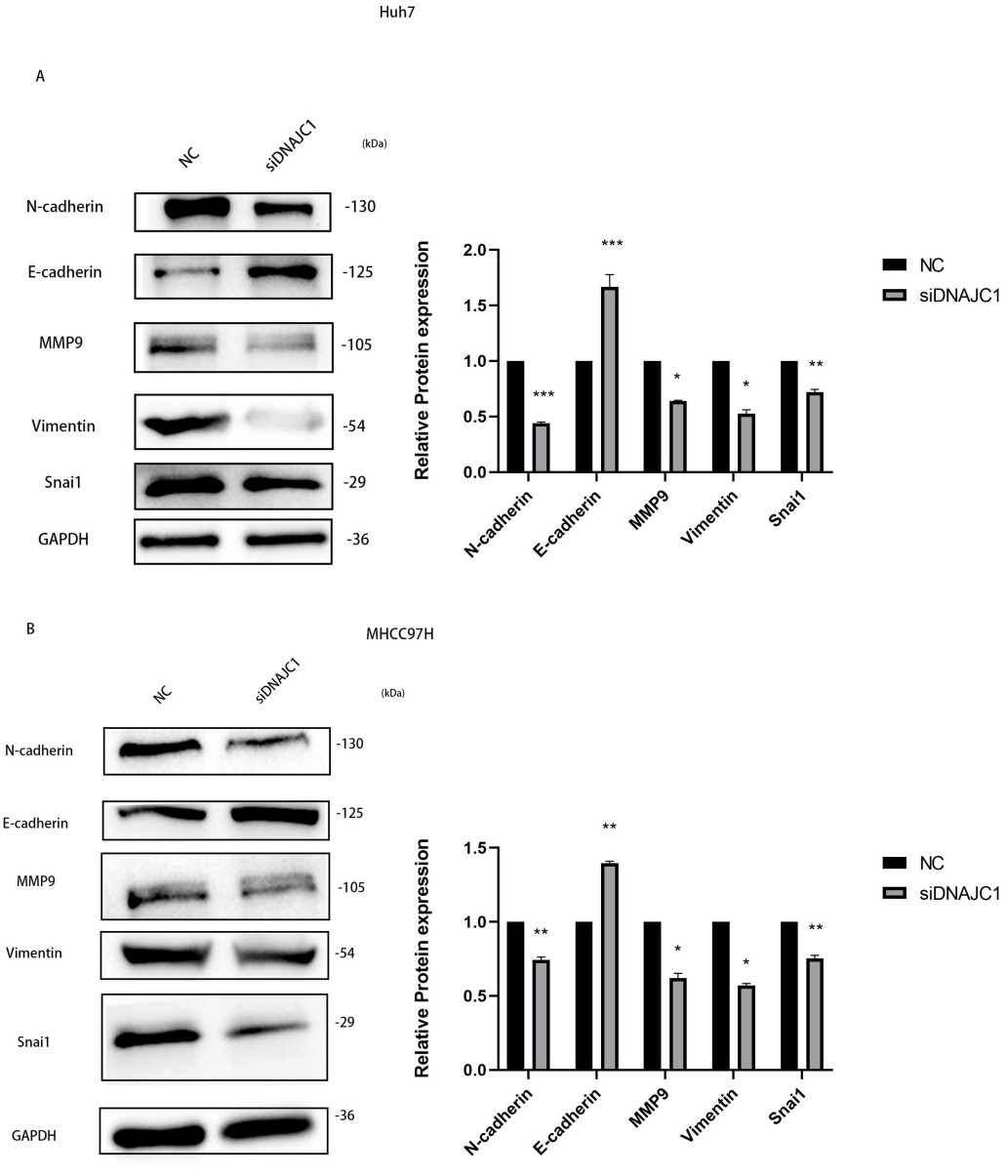

**Figure 9** **Silencing DNAJC1 promotes Huh7 and MHCC97H cells migration *via* EMT pathway.** (A) Silencing DNAJC1 promotes Huh7 cells migration *via* EMT pathway. (B) Silencing DNAJC1 promotes MHCC97H cells migration *via* EMT pathway. *: $p < 0.05$; **: $p < 0.01$; ***: $p < 0.001$.

and metabolism to play a role in cancer suppression (*Duffy et al., 2022*). Recent studies had shown that HSP40/J domain protein family members have regulatory effects on p53 and cancer signal transduction (*Kaida & Iwakuma, 2021*). However, the role of DNAJC1 in apoptosis in HCC cells was unclear according to the p53 pathway. Apoptosis is a spontaneous and orderly cell death controlled by genes, which is very important for maintaining the stability of the internal environment (*Bowen, 1993*). In the cell apoptosis process, the imbalance of pro-apoptotic proteins such as Bax and anti-apoptotic proteins

such as Bcl-2 induces increased mitochondrial permeability, leading to the disorder of mitochondrial permeability and the release of cytochrome C (*Brunelle & Letai, 2009*). In the study, we found that the upregulation of Bax and the downregulation of Bcl-2 after silencing DNAJC1, we speculated that silencing DNAJC1 might promote cells apoptosis. Furthermore, p21 plays an important role in apoptosis through p53-dependent and p53-independent pathways (*Abbas & Dutta, 2009*; *Qian & Chen, 2010*). p53 induces p21 expression in response to cellular stress, such as DNA damage (*Shamloo & Usluer, 2019*). In this investigation, compared to the NC group, the knockdown of DNAJC1 in HCC cells promoted the expression of the p21, p53 and p-p53(Ser20) proteins. Additionally, PARP-1 is important in DNA replication (*Hanzlikova et al., 2018*). Deactivation of PARP-1 can sensitize cancer cells to death by interfering with DNA repair and replication (*Wang, Luo & Wang, 2019*). Our research showed that the expression of PARP-1 significantly lower in siDNAJC1 group compared with the NC. These results further confirmed that silencing of DNAJC1 may promote apoptosis of HCC cells.

Epithelial–mesenchymal transition (EMT) was known to play an important role in cancer progression, metastasis (*Du & Shim, 2016*). EMT refers to various changes in cells at a molecular level, cells undergoing EMT display a decreased expression level of epithelial genes (such as E-cadherin) and an increased expression level of mesenchymal genes (such as N-cadherin, vimentin) (*Lamouille, Xu & Derynck, 2014*). In our study, we found that knockdown of DNAJC1 in HCC cells promoted the expression of E-cadherin and reduced the expression of N-cadherin and vimentin. Snai1 is an effector transcription factor, and it has been reported that Snai1 induces EMT by directly inhibiting E-cadherin transcription in tumors, thus promoting the growth, migration, and invasion of tumor cells (*Cano et al., 2000*; *Kudo-Saito et al., 2009*). In addition, studies have shown that matrix metalloproteinases (MMPs) can degrade extracellular matrix components and promote cancer cell invasion and metastasis especially the expression of MMP-9 is regulated by Snai1 (*Babaei, Aziz & Jaghi, 2021*). Our research showed that the expression of Snai1 and MMP9 was significantly lower in the siDNAJC1 group compared to the NC group. As expected, knockdown of DNAJC1 inhibited the HCC cells EMT.

## CONCLUSIONS

We demonstrated that DNAJC1 was highly expressed in HCC and associated with the prognosis of patients with HCC. DNAJC1 knockdown can inhibit cell proliferation, migration, invasion, and promote apoptosis. Additionally, DNAJC1 may mediate HCC cell apoptosis *via* p53 signaling pathway and mediate HCC cell migration *via* EMT signaling pathway.

**Abbreviations**

| | |
|---|---|
| **DNAJC1** | DNAJ Heat Shock Protein Family Member C1 |
| **HCC** | Hepatocellular Carcinoma |
| **IHC** | Immunohistochemical |
| **HSP40** | DNAJ heat shock protein family |
| **TIMER** | Tumor Immune Estimation Resource |

| | |
|---|---|
| **TCGA** | The Cancer Genome Atlas |
| **GSEA** | Gene Set Enrichment Analysis |
| **FBS** | Fetal Bovine Serum |
| **PVDF** | Polyvinylidene fluoride |
| **TBST** | Tris–HCl solution + Tween-20 |
| **NC** | Negative Control |
| **CCK-8** | Cell Counting Kit-8 |
| **PBS** | Phosphate Buffer Saline |

### Funding

This research was supported by the Research project on high-level talents of Youjiang Medical College for Nationalities (Grant No. YY2021SK02), the Foundation of Nanning Qingxiu District Key Research and Development Project (Grant No. 2020023), the Key Laboratory of Minimally Invasive Techniques & Rapid Rehabilitation of Digestive System Tumor of Zhejiang Province (Grant No. 21SZDSYS13) Open Fund from Key Laboratory of Cellular Physiology (Shanxi Medical University), Ministry of Education, China (Grant No. CELLPHYSIOL/SXMU-2021-08), the Grant of National-level project of university students' innovation and entrepreneurship in 2022 (Grant No. 202210599005; 202210599007) and the Grant of Guangxi provincial-level project of university students' innovation and entrepreneurship in 2022 (Grant No. S202210599056). The funders had no role in study design, data collection and analysis, decision to publish, or preparation of the manuscript.

### Grant Disclosures

The following grant information was disclosed by the authors:
Research project on high-level talents of Youjiang Medical College for Nationalities: YY2021SK02.
Foundation of Nanning Qingxiu District Key Research and Development Project: 2020023.
Key Laboratory of Minimally Invasive Techniques & Rapid Rehabilitation of Digestive System Tumor of Zhejiang Province: 21SZDSYS13.
Open Fund from Key Laboratory of Cellular Physiology (Shanxi Medical University).
Ministry of Education, China: CELLPHYSIOL/SXMU-2021-08.
Grant of National-level project of university students' innovation and entrepreneurship in 2022: 202210599005, 202210599007.
Grant of Guangxi provincial-level project of university students' innovation and entrepreneurship in 2022: S202210599056.

### Competing Interests

All authors declare that they have no financial, personal interests or beliefs that affect their objectivity, and have no economic or personal relationships with other people or organizations that improperly influence or prejudice their works.

## Author Contributions

- Yu-Chun Fan conceived and designed the experiments, performed the experiments, analyzed the data, prepared figures and/or tables, authored or reviewed drafts of the article, and approved the final draft.
- Zhi-Yong Meng conceived and designed the experiments, performed the experiments, analyzed the data, prepared figures and/or tables, authored or reviewed drafts of the article, and approved the final draft.
- Chao-Sheng Zhang performed the experiments, authored or reviewed drafts of the article, and approved the final draft.
- De-Wei Wei performed the experiments, authored or reviewed drafts of the article, and approved the final draft.
- Wan-Shuo Wei performed the experiments, authored or reviewed drafts of the article, and approved the final draft.
- Xian-Dong Xie performed the experiments, authored or reviewed drafts of the article, and approved the final draft.
- Ming-Lu Huang performed the experiments, authored or reviewed drafts of the article, and approved the final draft.
- Li-He Jiang conceived and designed the experiments, performed the experiments, analyzed the data, prepared figures and/or tables, authored or reviewed drafts of the article, and approved the final draft.

## Human Ethics

The following information was supplied relating to ethical approvals (i.e., approving body and any reference numbers):

All procedures were performed according to the Ethical Guidelines for Human Genome/Gene Research and were approved by the Ethics Committee of Affiliated Hospital of Youjiang Medical College for Nationalities (2022090501).

## Data Availability

The data is available at GitHub and Zenodo: https://github.com/Yu-ChunFan/Peer-J/tree/V1.1.2.

Yu-ChunFan. (2023). Yu-ChunFan/Peer-J: V1.1.2 (V1.1.2). Zenodo. https://doi.org/10.5281/zenodo.7962889

## Supplemental Information

Supplemental information for this article can be found online at http://dx.doi.org/10.7717/peerj.15700#supplemental-information.

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
