# Peer review of "DNAJ heat shock protein family member C1 can regulate proliferation and migration in hepatocellular carcinoma"

_PeerJ, doi:10.7717/peerj.15700_

## Round 0.1 · original submission · Major Revisions

Your manuscript was considered interesting by the reviewers however they had a number of significant and major concerns that need to be addressed. They expressed a concern about the relevance of the computational analyses you performed, the lack of detail about these analyses in the manuscript, and lastly that some of the results of these analyses provide weak evidence to support your hypothesis. Additionally, the reviewers stated there are several issues with the figures in the manuscript, including the legends and labels lacking sufficient explanation, as well as a concern that some of the figures need to be reorganized, addressed more clearly in the results section or perhaps even omitted from the manuscript because they provide weak evidence to support your results, which leads to an overstatement of your conclusions.

Please, submit a detailed rebuttal which shows where and how you have taken all comments and suggestions into consideration. If you do not agree with some of the reviewers’ comments or suggestions, please explain why. Your rebuttal will be critical in making a final decision on your manuscript. Please, note also that your revised version may enter a new round of review by the same or by different reviewers. Therefore, I cannot guarantee that your manuscript will eventually be accepted.

Reviewer 1 ·

Basic reporting

In the manuscript titled “DNAJ heat shock protein family member C1 can regulate proliferation and metastasis in hepatocellular carcinoma” the authors attempt to connect the elevated expression of DNAJC1 in hepatocellular carcinoma (HCC) with patient survival, cell proliferation, migration, and immune invasion of tumors. While the current manuscript is in a presentable format, it lacks a clear context as to why the authors pursued many correlation analyses and poorly explains and presents the data throughout. The figure legends throughout the paper are significantly lacking sufficient explanation for any reader to understand or assess the data. The results sections throughout the paper fail to thoroughly explain the data and gloss over many important pieces of information that would add context and improve understanding of the data. They attempt to make a connection to immune cell invasion but it’s unclear from the data if there is a coherent correlation with anything that would either positively or negatively impact HCC progression (this data is also explained in a way that made me feel misled about the data – line 230/231 grossly overstates what they find).

Fig 1A is showing the methylation sites along the DNAJC1 lacks any promoter/gene context that would allow a reader to consider the point the authors are trying to make. Figure 1B and C need to be organized more clearly so that readers could quickly assess the data (while some of the labels are obvious many are not, including HCC). Furthermore, this is presented in the text as “significantly” and “remarkably” altered when the figure shows a general increase in methylation/expression variability.

Figure 2 is showing expression analysis of DNAJC1 in normal vs HCC tumor. Figure legend is missing many useful pieces of information regarding statistical analyses especially in light of the visually unclear conclusion that they are drawing from it. A/B seem redundant, C shows DNAJC1 expression negatively correlates with age and D/E are visually unimpressive but lack any statistical explanation that might help understand the figure. The text states that DNAJC1 correlates with age and pathological stage of HCC patients which, given the previous figures conclusion (DNAJC1 expression is higher in HCC tumors), its confusing why they show contradictory evidence (age/maybe stage) without a clear explanation.

Figure 3 suffers from the same issues as all figures. 3A uses “altered” and “unaltered” group which is very confusing. 3F and G are too poorly described to assess what they are showing.

Figure 4 mentions the study of differentially expressed genes as differences between high and low DnaJC1 expression but this information is missing from the results section and create unnecessary confusion. I’m not sure why they are showing GO and KEGG analysis when many of the identified processes and pathways are not clearly associated with cancer progression. If I am wrong in this assessment, the authors need to more clearly state why they are presenting the information as support for DNAJC1 involvement in HCC. The GSEA plots show that high expression of DNAJC1 correlate with low p53, PI3K/AKT/mTOR, and Wnt/b-Catenin signaling. In general, p53 suppression is common for many cancers but mTOR/Wnt signalling are generally elevated (including for HCC). These plots show that high DNAJC1 correlates with low signaling something that is counterintuitive and unexplained in the text. The immune signaling plots make sense in the context of pursuing immune invasion correlations but are poorly described and ultimately inconclusive as presented in this manuscript.

Figure 5: While I am unfamiliar with TIMER correlations, I found many examples of much higher correlations in published manuscripts. This suggests to me that the correlation is weak even by the complicated calculations used by TIMER. 5C further confuses the point they are trying to make but showing that activated CD4 memory cells (a good sign for immune response to tumors is correlated with higher DNAJC1). Meanwhile resting CD4 memory cells and naïve B-cells (not sure of their importance in tumor immunogenicity) correlate with lower DNAJC1 expression. Furthermore, NK cells resting (positively correlated) is barely visible in 5B giving the impression of having the potential for high noise. If this is an inaccurate characterization of the data or its importance in HCC, there should be considerable explanation and context provided in the text. In its current form it provides more questions than answers.

Figure 6 has similar issues as figure 5. All correlations are very weak and the plots in 6B (with the minor exception of CCL15) show little observable connection (positively or negatively) with DNAJC1 expression. Furthermore, CCL15 expression was recently shown to negatively correlate with clinical outcome in HCC and may promote immune suppression in the tumor microenvironment (Liu LZ et al. 2019 Hepatology). There is no mention of this or how DNAJC1 might fit into this model.

Figures 7-10 present some of the strongest evidence in the manuscript but are not presented or discussed in terms of much of the first 6 figures (aside from elevated expression of DNAJC1 in HCC). Figure 7D has no figure legend. RNAi of DnaJC1 should also include L02 as a control to survey the general importance of DNAJC1 in “normal” cells for cell viability, migration, apoptosis. Notably, L02 migration has been studied before with these assays (Wang C. 2021 Frontiers Cell Infection Biology). Of principle concern is DNAJC1-KD behaves similarly in all cells in vitro, in which case, the importance of elevated DNAJC1 in HCC is more complicated than presented.

In conclusion, this paper offers a few interesting observations but fails to provide the context necessary to fully understand DNAJC1 impact on HCC progression. Furthermore, much of the bioinformatics is left unexplained, without context, and contradictory. It is the opinion of this reviewer that the manuscript requires a complete rewrite and consideration of the meaning of each figure in relation to HCC progression, immune cell activities, chemokine secretion, and HCC cellular viability. While some of the data is compelling, it is not suitable for publication in PeerJ.

Experimental design

This manuscript in its current form has many issues in data organization, methods description, figure description, and results/discussion. The authors would benefit from a careful analysis of how each figure fits together to tell the story of why DNAJC1 is important for HCC progression. I am convinced that DNAJC1 expression does offer some benefit to HCC progression but remain unconvinced that it has any meaningful impact on tumor immune cell invasion or chemokine expression. A more detailed analysis of DNAJC1 function in HCC would significantly improve this manuscript.

Validity of the findings

The authors effectively show DNAJC1 expression is elevated in HCC and seems to correlate with poorer outcomes and survival. However, many of the bioinformatic figures are presented with little consideration with the conclusions of the previous figures and are sometimes contradictory of one another. Figures 7-10 offer some good evidence supporting Figures 1 and 2 but need to be more fully analyzed and described to provide a coherent context to the study. Much of the figures 1-6 should be omitted and the remaining figures should be expanded on to better understand the importance of DNAJC1 expression in HCC. There are many instances where the correlations are poor at best and somehow have P values that are highly significant. The authors must significantly improve the explanation of the experiment, algorithm, and statistical output to provide a better understanding or omit it entirely.

·

Basic reporting

The manuscript titled “DNAJ heat shock protein family member C1 can regulate proliferation and metastasis in hepatocellular carcinoma” investigated the role of DNAJC1, a member of HSP40 family, in HCC. The authors first used bioinformatics tools to analysis the expression and potential effects of NDAJC1 in HCC, then performed experiments to see its effects after knockdown in HCC cells, confirming part of the bioinformatic results.
The idea that DNAJC1 has anti-cancer property and potential to be a prognostic maker is not novel. In general, HSPs including HSP40 all have these properties (please see PMID: 28012700, https://pubmed.ncbi.nlm.nih.gov/28012700/).
Overall, the English writing of the manuscript is OK. Here are some language issues or basic concepts that need to be revised:
1. In abstract, “Importantly, HCC cell proliferation, migration, invasion and apoptosis were
inhibited by …”, apoptosis won’t be inhibited.
2. Line 44, “Hepatocellular carcinoma (HCC) is the most common cancer in the world.” This statement is not correct. Do you mean mortality or incidence? Please see https://www.cancerresearchuk.org/health-professional/cancer-statistics/worldwide-cancer#heading-Zero.
3. Line 63, the statement “DNAJC1 has not been studied in HCC.” is not accurate. Please see https://www.ncbi.nlm.nih.gov/pmc/articles/PMC7457228/, https://www.mdpi.com/2079-7737/10/7/640, etc.
4. Line 74, de-capitalize “Methylation”.
5. Line 221, space out “Figure4D”.
6. Line 230, space out “Figure5A”

Experimental design

1. More information on interpretation of Figure 1 is needed. For example, Figure 1A is the distribution of DNAJC1 methylation sites. How can we tell from the figure? What do the numbers of x-axis mean?
Lines 191-192 says “methylation level of DNAJC1 in many tumor tissues was significantly lower than that in normal tissues”. In Figure 1B, based on the statistical label on the top, only 6 samples have significances vs. ns in 17 samples. Which sample is HCC? Need to point it out.
In Figure 1C, blue bar is normal tissues?
2. In Figure 2, how many patients were analyzed? In Figure 1A, what normal tissues were used?
3. In Figure 3A, what do blue and red lines stand for? The label is not clear. In Figure 3C, what is the significance of AUC? Need more explanation.
What’s the difference between Figure 3D and 3E?
Are Figure 3F and 3G incorrectly labelled? Not consistent with the description in manuscript (lines 205-207). Figure 3G is the nomogram to predict 1, 3, 5 survival, right? What is Figure 3F then?
4. What is the significance of Figure 4A-4C, any indication of DNACJ1 in HCC? If not, it makes the results complicated and irrelevant, and I suggest remove it or put it in supplemental. GSEA data should be enough.
5. Line 232, what are “gamma delta”?
6. Line 238, is LGALS9 considered as an immune checkpoint protein?
7. Figure 8C, huh7n cells, the inhibition curve didn’t show inhibition after siRNA knockdown, very different than MHCC97H cells.
8. Figure 10B, bar graph of relative protein expression, how the results are normalized? It showed normalized to GAPDH, NC and siRNA each normalized to its GAPDH, or all normalized to NC GAPDH. The bar graph is confusion, for example, why PARP expression is low? After DNAJC1 knockdown, apoptosis will increase, and PARP should be activated, right?
Figure 10B and 10C, need to label which cell is used.

Validity of the findings

Please see comments in section 2 “Experimental design”

Additional comments

No comment

Reviewer 3 ·

Basic reporting

In this study, the authors investigated the expression levels of DNAJC1 in different cancer types. They observed that DNJC1 was over expressed in cancer especially in HCC. Over expression of DNAJC1 could be driven hypomethylation. Authors showed that knockdown of DNAJC1 reduces proliferation, invasion, and migration. The strength of this manuscript is in vitro experiments data. I appreciate the authors for showing methylation data as part of bioinformatics data. However, a significant portion of bioinformatics data contains overstated claims and less clear. In addition, bioinformatics methods were not explained in detail. I suggest authors need to exercise due diligence while claiming the differences/effects they claim to find. I feel this manuscript would read well and be suitable for publication if the authors remove some problematic data in bioinformatic analysis.

Please address the following.

1. The results do not show DNJC1 regulate metastasis. DNAJC1 regulates proliferation, invasion, and migration. It is incorrect to assume that regulating invasion and migration is equivalent to regulating metastasis. Please correct the title, results, and discussion accordingly.
2. Line 79: How and why did you use logistic regression for this comparison? What is the predictor variable in these regressions? How was it dichotomized? Which figure shows this data? Please elaborate this analysis.
3. Line 71: What form of data was downloaded from TCGA? Read counts/FPKM/TPM/RSEM?
4. Line 89: Elaborate differential expression analysis? What was the input? What normalization? Is it log2FC or log10FC?
5. Line 107: What R software? Please elaborate these methods. What package was used?
6. What is X axis showing figure 1A?
7. Figure 1C: What is the rationale for dichotomizing the age at 65 years?
8. Figure 1D and E: Authors claimed that DNAJC1 expression changes with TNM stages and t stages. Most of these p values are insignificant. Median expression at these categories did not change in high direction with increasing stage/t stage. Authors claims are not supported by the data. It would be good to remove these two panels from the manuscript.
9. Figure 3F: The points allotted for each variable in the nomogram do not appear to be in sync with their coefficients in regression analysis. How was this nomogram constructed? Why variable importance is not following the trend in multivariate analysis? Please share the R code for nomogram construction.
10. Why were age, gender and grade used in nomogram? These variables were not independent predictors.
11. Figure 5 and 6: None of these correlations are existing. The most reliable measure for estimating correlation in the analysis is the correlation coefficient. All the correlations shown in this figure are close to zero indicating there is no real correlation. I ask authors to not to get mislead by the p value. It just explains the certainty of correlation coefficient. Altogether, the immune cell correlation and subsequent checkpoints and chemokines are misleading. I suggest that this paper will red well and accurate without these figures.
12. Figures 7, 8, 9 and 10: Please write (in corresponding figure legends) what p values are denoted by different asterisks marks in the figures.
13. Figure 8C: What is inhibition of infection? What do lines plots on the right show? There is no explanation of this in the text.

Experimental design

I have no comments on this.

Validity of the findings

I have no comments on this.

---

## Round 0.2 · Minor Revisions

Thank you for addressing the reviewers’ comments. There are a couple of additional concerns that need to be addressed. First, five new authors were introduced in the revised version of your manuscript, bringing the total number of authors from three to eight.

The author contribution section states that these authors edited and revised the manuscript, but does not list which of the original or new authors performed data collection for the experiments detailed in your manuscript (flow cytometry, western blotting, wound healing, or any of the other assays). Since the flow cytometry section was added to the revised manuscript, were the new authors responsible for these experiments and the corresponding section of the manuscript? It needs to be specified which authors performed data collection for specific experimental sections.

Additionally, there is a concern that not all the raw blot images used to prepare the manuscript figures have been deposited in your Zenodo repository. Could you please address and correct this discrepancy?

In addition, the manuscript would greatly benefit from professional copy-editing.

Please, submit a detailed rebuttal that shows where and how you have taken all comments and suggestions into consideration. If you do not agree with some of the reviewers’ comments or suggestions, please explain why. Your rebuttal will be critical in making a final decision on your manuscript. Please, note also that your revised version may enter a new round of review by the same or by different reviewers. Therefore, I cannot guarantee that your manuscript will eventually be accepted.

·

Basic reporting

The authors addressed reviewers’ comments well and revised the manuscript based on the comments. The quality of revised manuscript has improved.

Experimental design

The authors updated the experimental design based on reviewers’ comments.

Validity of the findings

The authors updated the results based on reviewers’ comments.

Reviewer 3 ·

Basic reporting

Authors addressed my comments.

Experimental design

None

Validity of the findings

None

---

## Round 0.3 · accepted · Accept

All remaining concerns of the reviewers were adequately addressed and revised manuscript is acceptable now.

·

Basic reporting

Good and acceptable.

Experimental design

Good and acceptable.

Validity of the findings

Good and acceptable.

Additional comments

No.

Reviewer 3 ·

Basic reporting

The authors addressed my concerns. I have no further comments.

Experimental design

The authors addressed my concerns. I have no further comments.

Validity of the findings

The authors addressed my concerns. I have no further comments.